# Using a 3D Navigation Template to Increase the Accuracy of Thoracic Pedicle Screws in Patients with Scoliosis

**DOI:** 10.3390/bioengineering10070756

**Published:** 2023-06-25

**Authors:** Cheng-Hao Jiang, Yan Shi, Yong-Mei Sun, Ming-Jun Cai, Hai-Long Wu, Li-Sheng Hu, Li-Min Yu, Peng Wang, Jie Shen, Yong-Can Huang, Bin-Sheng Yu

**Affiliations:** 1Clinical College, Peking University Shenzhen Hospital, Anhui Medical University, Shenzhen 518036, China; jch_zurich@163.com (C.-H.J.);; 2Shenzhen Key Laboratory of Spine Surgery, Department of Spine Surgery, Peking University Shenzhen Hospital, Shenzhen 518036, China; 3Institute of Orthopaedics, Shenzhen Peking University-Hong Kong University of Science and Technology Medical Center, Shenzhen 518036, China; 4The Fifth Clinical Medical College, Anhui Medical University, Hefei 230032, China; 5Shenzhen Engineering Laboratory of Orthopaedic Regenerative Technologies, National & Local Joint Engineering Research Center of Orthopaedic Biomaterials, Peking University Shenzhen Hospital, Shenzhen 518036, China

**Keywords:** pedicle screw placement, navigation template, accuracy, scoliosis, rotation

## Abstract

This study compares the accuracy and safety of pedicle screw placement using a 3D navigation template with the free-hand fluoroscopy technique in scoliotic patients. Fifteen scoliotic patients were recruited and divided into a template group (eight cases) and a free-hand group (seven cases). All patients received posterior corrective surgeries, and the pedicle screw was placed using a 3D navigation template or a free-hand technique. After surgery, the positions of the pedicle screws were evaluated using CT. A total of 264 pedicle screws were implanted in 15 patients. Both the two techniques were found to achieve satisfactory safety of screw insertion in scoliotic patients (89.9% vs. 90.5%). In the thoracic region, the 3D navigation template was able to achieve a much higher accuracy of screw than the free-hand technique (75.3% vs. 60.4%). In the two groups, the accuracy rates on the convex side were slightly higher than on the concave side, while no significance was seen. In terms of rotational vertebrae, no significant differences were seen in Grades I or II vertebrae between the two groups. In conclusion, the 3D navigation template technique significantly increased the accuracy of thoracic pedicle screw placement, which held great potential for extensively clinical application.

## 1. Introduction

Scoliosis is a complex, three-dimensional deformity of the spine with lateral deviation in the coronal plane (>10°), the alternation of kyphosis or lordosis in the sagittal plane and the rotation of the vertebrae in the axial plane [1]. The major coronal curve and axial vertebral rotation are two major factors that manifest the severity of spinal deformity, evaluate the risks of progression and predict the prognosis [2]. For severe scoliosis, correction surgery should be performed to ameliorate progressive cosmetic deformity, cardiopulmonary system and psychological distress; currently, posterior spinal pedicle screw instrumentation is the most commonly used technique of surgical treatment [3]. Several studies have confirmed that pedicle screw fixation provides the most powerful biomechanical strength via three-column fixation and achieves three-dimensional correction of scoliotic deformity [4,5,6]. The free-hand technique is currently the most common method for pedicle screw insertion, but the malposition of the pedicle screw occurs at a rate of 24.4–56.3% [7,8,9,10,11]; pedicle screw insertion in the scoliotic spine is much more challenging than that in the normal spine due to anatomical variations, smaller pedicle diameters and vertebral rotations [1,12,13]. Pedicle screw malposition can lead to treatment inefficacy and may cause severe damage to adjacent neurological and vascular structures [14,15,16,17]. To improve the accuracy, several techniques were applied to assist pedicle screw insertion, including intraoperative computer-assisted navigation, robot-assisted technique and 3D navigation templates.

The use of the 3D navigation template technique was first proposed in the 1990s [18], and the clinical application of templates has recently become more popular in the surgical treatment of different spine disorders [19]. Several studies have demonstrated that the 3D navigation templates improve the accuracy of screw placement, reduce the time taken for screw insertion and lower the frequency of intraoperative radiographs in scoliotic corrective surgeries [20,21,22]. Nevertheless, how 3D navigation templates affect rotational vertebrae is not well understood.

Scoliosis typically occurs more often in the thoracic spine than in the other spinal regions. Previous studies have suggested that transpedicular screw fixation may not be suitable for most midthoracic regions because of the smaller pedicles and adjacent vital structures [23,24]; hence, pedicle screw insertion in the thoracic region is technically challenging. Additionally, the vertebrae in the scoliotic spine rotate towards the convex side with the shift of the spinal cord to the concave side; thus, the width of the epidural space on the concave side is significantly smaller than that on the convex side in most levels [25]. Hence, surgeons need to pay greater attention to pedicle screw insertion on the concave side to avoid neurological injury during scoliosis correction. Axial vertebral rotation is an essential feature of scoliotic deformity and directly manifests the severity of the deformity [26]. The goal of the surgical treatment of scoliosis is to achieve a three-dimensional correction of deformity, and axial vertebral derotation improves rib hump deformity, cardiopulmonary function and scapular prominence [27,28]. Additionally, the derotation of vertebrae is associated with long-term positive follow-up outcomes [2,29]. Thus, it is essential to determine whether the use of 3D navigation templates can improve the accuracy of pedicle screw insertion in the rotational vertebrae of the spine.

Therefore, as mentioned above, the main aim of this retrospective comparative analysis was to investigate the accuracy and safety of pedicle screw insertion using a 3D navigation template and the free-hand technique to correct the scoliotic spine, especially in the thoracic region. The second aim of this study was to investigate the difference in the accuracy of pedicle screw insertion with 3D navigation templates between the concave side and convex side in scoliosis correction surgery, and the third aim was to determine the accuracy of pedicle screw insertion with 3D navigation templates in vertebrae with different rotational angles.

## 2. Materials and Methods

### 2.1. Research Ethics and Patients’ Section

This study was approved by the Research Ethics Committee of Peking University Shenzhen Hospital (2022-041) and was carried out in accordance with the Code of Ethics of the World Medical Association (Declaration of Helsinki). Additionally, this study was registered at Chinese Clinical Trial Registry (www.chictr.org.cn, accessed on 7 April 2023) with the registration number ChiCTR2300070277. All patients signed informed consent forms for the operation. A retrospective study was performed to analyse the data obtained for the scoliotic patients who underwent surgical treatment between January 2014 and December 2021. Inclusion criteria: (1) patients diagnosed with scoliosis who underwent primary posterior correction surgery, (2) the pedicle screw insertion was carried out using the free-hand technique or the 3D navigation template technique and (3) the postoperative data were complete. Exclusion criteria: (1) patients who received revision surgery or combined anterior–posterior surgery, (2) the pedicle screws were placed via other techniques and (3) other spinal surgeries were performed simultaneously.

### 2.2. Preoperative Measurements

The patients’ preoperative spinal radiographs and standard-dose 3D CT scans (0.625 mm slice thickness) were used for evaluation and measurement. As illustrated in Figure 1, the measurements of transverse pedicle width and the pedicle–rib unit (PRU) width were adopted using O’Brien’s method [30]. Vertebral rotation was accessed and graded using Nash–Moe criteria on preoperative anteroposterior radiographs [31]. Two spine surgeons who were not involved in this study were invited to measure the rotation, pedicle width and PRU width blindly and independently. In terms of vertebral rotation, differences were resolved by consensuses with the participation of another spine surgeon.

### 2.3. Fabrication of the Spine Model and 3D Navigation Templates

The patients’ CT data were stored in DICOM format and imported into Mimics software (Materialise, Leuven, Belgium) to generate a 3D reconstruction spine model. The optimal pedicle screw entry point and trajectory were designed with the aim of avoiding any penetration of the wall of the pedicle or vertebrae. The model (STL format) was then imported into 3-Matics software to design the matched virtual 3D navigation templates; the exported file of the 3D navigation template 3-Matics design results was imported into the 3D printer and the 3D navigation template was printed on a 3D printer (Stratesys, Eden260V, Rehovot, Israel) provided by Shenzhen Excellent Technology Company Research Institution using photosensitive polyamide (layer thickness = 16 um). As shown in Figure 2, the 3D navigation template comprised three main parts, namely two substrates (drilling cannula) and a connecting bar, and the drilling cannula was a hollowed column with an inner diameter of 2.5 mm to allow a 2.0 mm drill bit to pass through. The surgeons used the connecting bar to press and stabilise the 3D navigation template and to avoid the slippage or tilting of the template on the vertebra. Before surgery, a 3D navigation template was placed on the spine model to assess its fitness and stability, and the templates were sterilised before use by a low-temperature plasma sterilisation method.

### 2.4. Surgical Procedures and the Use of 3D Navigation Templates

All surgeries were performed using the posterior surgical approach after general anaesthesia. For congenital scoliosis, hemivertebra resection and short segment fusion were performed in all patients. In the thoracic region, due to the anatomical variance of pedicels, transpedicular screw fixation may not be suitable for most thoracic pedicles. An alternative method involving extrapedicular screw fixation within the PRU was evaluated and found to be anatomically feasible [32]. Although the PRU could not provide the same transpedicular fixation strength as the pedicles, it met the basic biomechanical needs of the screw–rod system [33].

In the free-hand group, the entry point and direction were confirmed by anatomical landmarks and perioperative radiographs. After entering and widening the pedicle with an awl, the trajectory hole was explored with a ball-tipped probe to verify the integrity of the pedicular walls and exclude any cortical penetration. After the placement of the pedicle screw, anteroposterior and lateral fluoroscopies were performed to confirm the position of the screw in the free-hand group if necessary. In the template group, after removing the soft tissues surrounding the facet joint and lamina, the 3D navigation templates were tightly attached to the posterior bone surface of the laminae and upper articular processes. The surgeons used an electric bar to prepare the screw trajectory with the guidance of templates. During the drilling procedure, the surgeons pressed the connecting bar to keep the template stable, and the pedicle trajectory was enlarged using an awl before a pedicle screw was inserted along the trajectory, fluoroscopies were carried out after all screws were placed. The postoperative treatments and rehabilitation were similar between the two groups.

### 2.5. Methods of Pedicle Screw Grading

The radiographs of the whole spine and a standard-dose 3D CT scan for each patient were performed postoperatively, and all pedicle screws were evaluated using PACS (Picture Archiving and Communication Systems) software. As described previously [34], the position of pedicle screws was divided into four levels, according to the degree of perforation: Grade 0, no perforation; Grade 1, ≤2 mm; Grade 2, 2~4 mm; and Grade 3, >4 mm or screw complications occurred. For the thoracic spine, we adopted the extrapedicular screw fixation method within the PRU. Therefore, we regarded the PRU as the basis for grading screws (not pedicles), which is more reliable and credible. As shown in Figure 3, a pedicle screw that was completely within the PRU but not in the pedicle was also regarded as Grade 0. A pedicle screw placed within the pedicle or PRU was considered accurate, and a systematic review suggested that the perforation of the bone cortex of less than 2 mm was considered safe, whereas perforation of more than 2 mm was considered unsafe [35]. We classified the direction of screw perforation according to CT multiplanar reconstruction as follows: L, lateral side of pedicle or PRU; M, medial side of pedicle; A, anterolateral of vertebral column. Two spine surgeons who were not involved in this study were invited to evaluate all pedicle screws from 15 patients blindly and independently. Differences were resolved by consensus with the participation of a senior spine surgeon.

### 2.6. Statistical Analysis

Statistical analysis was performed using SPSS Statistics 25.0 software (IBM, USA). All continuous variables are presented as the mean and standard deviation. A two-tailed Student’s *t*-test was used to compare the preoperative measurements. Furthermore, the comparisons of accuracy and safety between the two groups were made using the χ^2^ test (the chi-squared test). *p*-values less than 0.05 were considered significant.

## 3. Results

Fifteen patients were recruited in this study (fourteen females and one male; mean age was 28.7 ± 15.1 years old), and Table 1 lists their clinical characteristics. As shown in Table 2, no statistical difference was seen between the two groups in terms of transverse pedicle width or PRU width on preoperative images. In the template group, the transverse pedicle width and PRU width on the concave side were 6.1 ± 2.5 mm and 13.0 ± 2.1 mm, respectively, and on the convex side, the values were 6.1 ± 2.6 mm and 12.4 ± 2.1 mm, respectively. In the free-hand group, the transverse pedicle width and PRU width on the concave side were 6.3 ± 2.5 mm and 12.4 ± 2.3 mm, respectively; on the convex side, the values were 6.2 ± 2.7 mm and 11.6 ± 3.3 mm, respectively. There was no obvious difference in transverse pedicle width or PRU width between the concave side and convex side in these two groups.

In total, 192 thoracic pedicles were implanted in 15 patients, and 49.5% of these pedicles had a transverse diameter less than 5.0 mm, which was not suitable for screw insertion because of insufficient space. In the template group, the transverse PRU width in middle thoracic spine (T5-T8) and lower thoracic spine (T9-T12) were 11.5 ± 1.5 mm (*n* = 38) and 14.2 ± 1.8 mm (*n* = 48), respectively; and in the free-hand group, the values were 11.2 ± 2.1 mm (*n* = 31) and 13.6 ± 2.0 mm (*n* = 41), respectively. In the two groups, the transverse PRU width in lower thoracic spine was larger than it in middle thoracic spine (*p*< 0.05). The numbers of vertebrae with different rotations were recorded according to the Nash–Moe method. The number of vertebrae at Grades I and II were 39 and 10 in the template group, respectively, and 21 and 15 in the free-hand group, respectively (Table 3).

Table 4 summarises the distribution of pedicle screws: 264 screws were implanted and the overall accuracy and safety in the template group were similar to that in the free-hand group (accuracy: 70.3% vs. 61.9%; safety: 89.9% vs. 90.5%). Nevertheless, in the thoracic region, the accuracy of the pedicle screws in the template group (75.3%) was much higher than that in the free-hand group (60.4%; *p* = 0.028). The accuracy of the pedicle screw on the convex side was higher than that of the concave side in the template group or free-hand group, but neither showed a statistical difference (template: 77.3% vs. 63.9%; free-hand: 69.8% vs. 54.0%). In the template group, 38 screws were placed in the middle thoracic spine (T5–T8) and 49 screws were placed in the lower thoracic spine (T9–T12), compared to 31 and 41 in the free-hand group, respectively; the accuracy in the lower thoracic spine was higher than in the middle thoracic spine (template: 71.1% vs. 83.7%; free-hand: 61.3% vs. 70.7%), but no significant differences were observed. As shown in Table 5, in terms of rotational vertebrae, the accuracy in Grades I and II were 76.1% and 55.0% in the template group, respectively, and 69.2% and 56.0% in the free-hand group, respectively. Although it seemed that there was a negative correlation between accuracy and rotation angle, we found that there were no significant differences between Grades I and II in the template group or free-hand group. In addition, the differences in accuracy in Grade I or Grade II between the two groups were not significant.

The most commonly reported complication was screw malposition. There were 89 perforations in the two groups (template vs. free-hand: 41 vs. 48); most frequently, these were medial perforations (template vs. free-hand: 26 vs. 28), followed by anterolateral perforations of the cortex of the vertebral body (template vs. free-hand: 7 vs. 15) and lateral perforations (template vs. free-hand: 4 vs. 4). Other perforations included four lateral-medial perforations and one inferior perforation. In the template group, two patients had CSF (cerebrospinal fluid) leakage and one had a malpositioned screw (free-hand group) with asymptomatic aortic abutment, but screw revision was not necessary.

As shown in Table 6, there was no significant difference in drainage volume between the two groups (*p* > 0.05). Additionally, no significant difference in postoperative hospital stay was observed between the two groups (*p* = 0.696; 15.4 ± 5.4 d vs. 14.1 ± 6.8 d).

As illustrated in Figure 4, the 3D navigation template technique was applied in a female patient with adult idiopathic scoliosis who underwent corrective surgery; after twenty-one screws were placed with 3D navigation templates (T4-L4), the trunk balance was obtained and the correction rate of Cobb angle was 50.6% postoperatively.

## 4. Discussion

Pedicle screw insertion is a challenging procedure in scoliosis correction surgery and misplacement of the screw may cause severe complications. This study investigated the accuracy and safety rate of the 3D navigation template in scoliosis surgery and found that this technique enhanced the accuracy of the pedicle screw in the thoracic region compared to the free-hand technique. Additionally, both 3D navigation template and free-hand technique were able to achieve high safety rates of pedicle screws during scoliosis correction surgery. According to the above results, the 3D navigation template has promising potential for improving the accuracy of pedicle screws in scoliotic corrections and thus deserves extensive application in clinics.

Due to the anatomic variations of deformed spines, pedicle screw misplacement is more likely to occur during corrective surgery [36]. Intraoperative computer-assisted navigation and the robot-assisted technique can improve the accuracy of pedicle screw placement compared to the free-hand technique; however, certain disadvantages of the two techniques limit their clinical application. The computer-assisted navigation technique is used by a minority of surgeons due to the enormous economical and biologic costs and space requirements in the operating room; it is also fault-prone and time-consuming, especially during the learning curve [37]. The robot-assisted technique can increase the accuracy of screws and reduce intraoperative radiation exposure time and dose [38,39]. However, this technique is expensive and has a long learning curve, which may not be practical for hospitals with only a few spinal instrumental surgeries each year.

The 3D navigation template technique, which has mainly been used in recent spinal corrective surgeries, improves the accuracy of pedicle screw placement based on clinical and cadaveric studies [20,40,41]. A previous comparative study found a significant difference between the template group and the free-hand group in the accuracy of pedicle screw placement (91.2% vs. 82.6%) [21]. In that study, five Grade 3 screws in two groups were recorded, which were not seen in our study; additionally, eleven congenital scoliotic patients and two kyphotic patients were recruited, and the difference in samples might lead to the discrepancy of the results between the study and ours. This is because most patients with congenital scoliosis need to receive corrective surgeries at a young age when pedicle screw placement is much more challenging because of the smaller pedicles. Cao and colleagues reported an excellent accuracy rate of pedicle screw placement with 3D navigation templates in congenital scoliosis, which was much higher than that of the free-hand technique (96.10% vs. 88.64%, *p* = 0.007) [42], the accuracy rates were much higher than ours, while the average number of screws per patient was lower.

Axial vertebral rotation is an important part of deformity in the scoliotic spine, which makes pedicles converge and increases the difficulty of placing screws precisely. A previous study confirmed that as the rotation angle increased, the accuracy of the pedicle screw declined in the lumbar spine [43]. In this study, we accessed the rotation angle of vertebrae on preoperative radiographs according to the Nash–Moe criteria. We compared the accuracy of screws between Grades I and II in the two groups, but no significant difference was seen. Moreover, statistical differences in accuracy in Grade I or Grade II were not seen between the 3D navigation template and free-hand techniques in this study. The elaborative preoperative plan, excellent surgical experience and smaller sample size might have contributed to this result; moreover, the rotation angles of Grade I or II were not too high [44], which has a limited impact on accuracy. In addition, the revisions of misplaced screws during operation might also reduce the negative impact of rotation on screw insertion.

The spinal cord always remains tethered on the concave side, even during correction, and a slightly malpositioned screw may have a catastrophic consequence on the neurologic outcome; thus, much more attention must be paid when the pedicle screw is placed on the concave side. For instance, a previous study with a sample of 976 pedicle screws observed that more misplaced pedicle screws were seen on the concave side in scoliosis correction (33.5% vs. 21.9%, *p* < 0.05) [45]. In this study, we observed that the accuracy of the pedicle screw was higher on the convex side in the two groups, but a significant difference was not seen between the concave and convex sides. The 3D navigation template technique might reduce the perforation rate of the pedicle screw due to its precision and specificity. However, a comparative study confirmed that there was a deviation between intraoperative actual screw trajectory and ideal trajectory designed preoperatively [46].

In this study, the malposition rates were 29.7% and 38.1% in template group and free-hand group, respectively; the deformed pedicles, residual soft tissue, intervertebral movements during screw insertion might lead to this result. Nevertheless, the safety rates were high in two groups, and no postoperative revisions were necessary. After surgery, two cases of CSF leakage were found in the template group and the drainage tubes were removed after conservative treatments; CSF leakage was not found during surgery and the Ponte osteotomy in severe idiopathic scoliosis and a hemivertebrae resection in congenital scoliosis might contribute to the delayed CSF leakage.

The 3D navigation template has many other advantages. A previous study confirmed that template technique was able to shorten the mean time for pedicle insertion and total surgical time, the mean time for one pedicle screw insertion in template group and in free-hand group were 6 min and 7 min, respectively [22]. Intraoperative radiation exposure is another issue that surgeons have to encounter, which may lead to deleterious healthy effects, such as malignancies [47]. Cecchinato and colleagues recorded the number of fluoroscopy shots and found that the use of navigation template could reduce the radiation dose greatly (template vs. free-hand: 11 shots vs. 47.5 shots) [20]. In this study, the surgeons did not strictly record the number of shots, but perioperative fluoroscopies were carried out in the free-hand group when the surgeons could not guarantee the accuracy of the screw; in the template group, the fluoroscopy was only performed after all the screws had been placed.

The design of the template should be optimised based on surgical and mechanical considerations to reduce soft tissue removal but not sacrifice stability. In our study, the 3D navigation template was designed with two substrates and a connecting bar, which did not increase the contact area and allowed surgeons to hold it to maintain the stability of the template. Moreover, preoperative and postoperative CT scans mean more radiation exposure to patients, a low-dose CT scan might be a better choice to reduce radiation exposure. In addition, the cost of spine model was about USD 400 per model, and the cost of the navigation template was about USD 105 per template, which were covered by our research fund. Nevertheless, we should acknowledge that there are several limitations to this study. First, the sample size was too small and the basic characteristics of the patients in two groups did not match well; the difference in accuracy between the navigation template and the free-hand technique could be more evident if a larger randomized controlled trial (RCT) could be conducted in the future. Furthermore, the distribution of screws on the concave and convex sides, or in different rotational vertebrae, should also be investigated. Third, surgeons did not record the radiation exposure and mean screw insertion time during surgery, so that we could not truly describe whether the using of the 3D navigation template decreased the radiation exposure and shortened the screw insertion time. Last but not least, we did not examine the follow-up outcomes of the included patients as this was not our major objective.

## 5. Conclusions

In this retrospective study, both techniques were found to achieve the satisfactory safety of screw insertion in scoliotic patients, and the 3D navigation template was demonstrated to increase the accuracy of pedicle screw placement in the thoracic region. Notably, screw malposition was more likely to occur on the concave side or in rotational pedicles; thus, more attention needed to be paid during screw insertion. In future studies, a larger RCT would help to further identify the value of the 3D navigation template in scoliotic corrections.

## Figures and Tables

**Figure 1 bioengineering-10-00756-f001:**
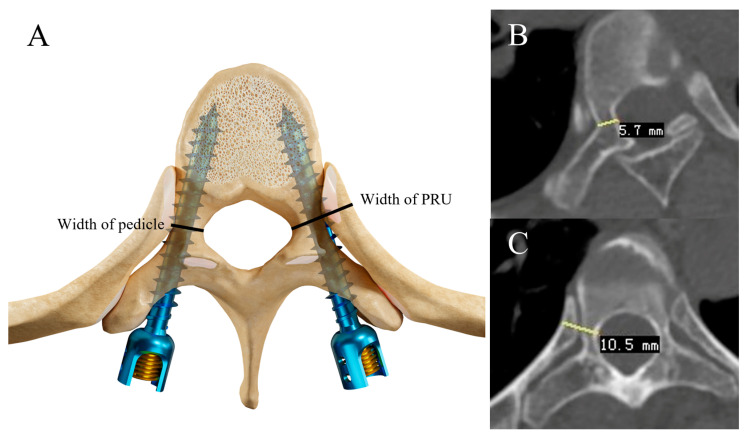
Schematic diagram showing measurement methods for transverse pedicle width, PRU width: (**A**) is the schematic diagrams demonstrating the measurements of pedicle and PRU width. Schematic diagrams (**B**,**C**) show the measurements we made using PACS.

**Figure 2 bioengineering-10-00756-f002:**
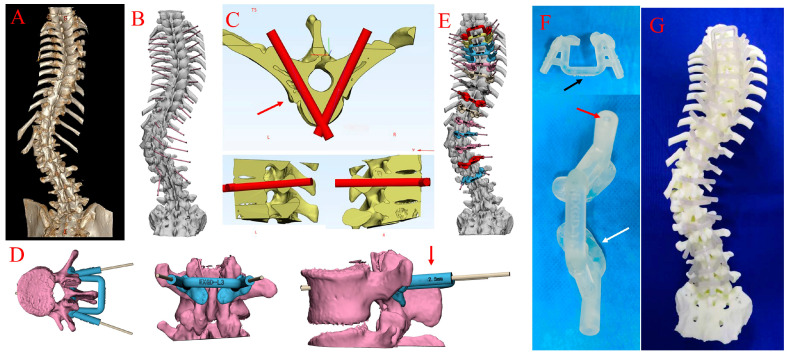
A schematic diagram of the design spine model and 3D navigation template. (**A**) Preoperative 3D CT reconstruction of the spine. (**B**) The 3D reconstruction of the spine using Mimics software. (**C**) Design an ideal virtue trajectory without any perforations. (**D**) The virtue navigation template with two drilling cannulas matching the vertebra well. The drilling cannula has an inner diameter of 2.5 mm (red arrow). (**E**) The 3D reconstruction of spine and virtue templates. (**F**) The physical 3D navigation template comprises three main components: two substrates (white arrow) and a connecting bar (black arrow). (**G**) The 3D printing spine model and 3D navigation templates.

**Figure 3 bioengineering-10-00756-f003:**
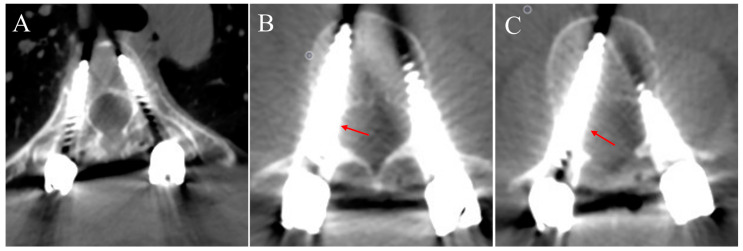
Illustrations of different grades of pedicle screws. (**A**) Grade 0: The screw was totally within the PRU. (**B**) Grade 1: The perforation of the medial wall was less than 2 mm (red arrow). (**C**) Grade 2: The perforation of the medial wall was more than 2 mm but less than 4 mm (red arrow).

**Figure 4 bioengineering-10-00756-f004:**
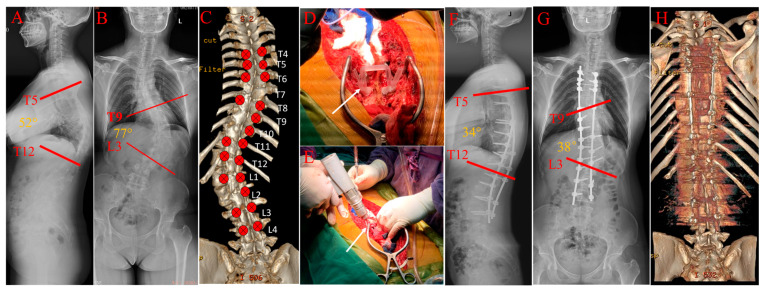
A case illustrating the use of 3D navigation template technique in scoliotic patient: A 28-year-old patient: (**A**–**C**): Preoperative X-ray and 3D-reconstruction CT. (**D**,**E**): Perioperative images of navigation templates (white arrow). (**F**–**H**): Postoperative X-ray and 3D-reconstruction CT.

**Table 1 bioengineering-10-00756-t001:** Basic characteristics of the patients.

	Template Group	Free-Hand Group
Number	8	7
Age (Year)	27.8 ± 12.8	29.9 ± 18.4
Gender(M/F)	(0/8)	(1/6)
Diagnosis	Idiopathic scoliosis (5 cases)	Idiopathic scoliosis (5 cases)
	Congenital scoliosis (3 cases)	Degenerative scoliosis (2 cases)
Mean Cobb Angle (°)	61 ± 11°	56 ± 17° (* *n* = 6)

* *n* = 6: one patient’s preoperative standing anteroposterior radiograph was missing.

**Table 2 bioengineering-10-00756-t002:** Measurements of pedicle width and PRU in two groups.

		Pedicle Number	Mean Pedicle Width (mm)	Mean PRU Width (mm)
Template		138	6.1 ± 2.6	-
Free-hand		126	6.2 ± 2.6	-
Thoracic	Template	101	5.5 ± 2.1	12.7 ± 2.1 (* *n* = 100)
	Free-hand	91	5.5 ± 2.0	12.4 ± 2.3 (*n* = 91)
Concave	Template	72	6.1 ± 2.5	13.0 ± 2.1
	Free-hand	63	6.3 ± 2.5	12.4 ± 2.3
Convex	Template	66	6.1 ± 2.6	12.4 ± 2.1
	Free-hand	63	6.2 ± 2.7	11.6 ± 3.3

* *n* =100: one patient lacked the twelfth rib.

**Table 3 bioengineering-10-00756-t003:** Number of vertebrae with different rotation angles in the two groups *.

	Grade 0	Grade I	Grade II	Grade III	Grade IV	Total
Free-hand (** *n* = 6)	18	21	15	1	0	55
Template	20	39	10	4	1	74

* The rotational vertebrae were graded according to the Nash–Moe method. ** *n* = 6: one patient’s preoperative standing anteroposterior radiograph was missing.

**Table 4 bioengineering-10-00756-t004:** Distribution of pedicle screws *.

		G 0	G 1	G 2	G 3	Total	Accuracy	Safety
Total	Template	97	27	14	0	138	70.3%	89.9%
	Free-hand	78	36	12	0	126	61.9%	90.5%
Thoracic	Template	76	16	9	0	101	75.3%	91.1%
	Free-hand	55	27	9	0	91	60.4%	90.1%
Concave	Template	46	20	6	0	72	63.9%	91.7%
	Free-hand	34	19	10	0	63	54.0%	84.1%
Convex	Template	51	7	8	0	65	77.3%	89.2%
	Free-hand	44	17	2	0	63	69.8%	96.8%

* The screw position was divided into four grades according to Mobbs–Raley method.

**Table 5 bioengineering-10-00756-t005:** Distribution of pedicle screws in different rotational vertebrae.

	Free-Hand		Template	
Rotation Classification	Grade I	Grade II	Grade I	Grade II
Accuracy	69.2%	56.0%	76.1%	55.0%

**Table 6 bioengineering-10-00756-t006:** Postoperative situation.

	Template	Free-Hand
Number of vertebrae	74	69
Drainage volume (mL)	1027.9 ± 414.2	982.4 ± 225.4
Complications	CSF leak (2 cases)	None
Postoperative hospital stay (day)	15.4 ± 5.1	14.1 ± 6.8

## Data Availability

Not available.

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
