# Peer review of "Using a 3D Navigation Template to Increase the Accuracy of Thoracic Pedicle Screws in Patients with Scoliosis"

_bioengineering, 2023, doi:10.3390/bioengineering10070756_

Round 1
Reviewer 1 Report
The study entitled “Using a 3D navigation template to increase the accuracy of thoracic pedicle screws in patients with scoliosis” aimed to compare the accuracy and safety of pedicle screw placement using a 3D navigation template versus the free-hand fluoroscopy technique in scoliotic patients. The study found that the 3D navigation template technique significantly increased the accuracy of thoracic pedicle screw placement compared to the free-hand technique. However, both techniques were found to achieve satisfactory safety of screw insertion in scoliotic patients.
Comments and Questions:
1. CSF leakage during scoliosis surgery can be a serious complication, which may be resulted from violence into nerve tissue and cause terrible neurologic deficits. Could you provide more clinical information about the two cases of CSF leakage during scoliosis surgery and explain the potential reasons why this complication occurred specifically in the 3D navigation template group and not in the free-hand technique group?
2. You may add footnotes to the tables in order to provide clear definitions for the classifications used in them, such as vertebral rotation and screw position, even if they have already been defined in the text.
3. The small sample size limited the accuracy. If the sample number could get more, it would be more powerful to convince readers.
4. I believe that the screw accuracy using the 3D template technique should have been superior to the free-hand technique in more rotatory vertebrae, but your study was not able to demonstrate this. Besides the small sample size, what other factors do you think may have contributed to the lack of significant difference in screw accuracy between the 3D template and free-hand techniques in more rotatory vertebrae in your study?
The study entitled “Using a 3D navigation template to increase the accuracy of thoracic pedicle screws in patients with scoliosis” aimed to compare the accuracy and safety of pedicle screw placement using a 3D navigation template versus the free-hand fluoroscopy technique in scoliotic patients. The study found that the 3D navigation template technique significantly increased the accuracy of thoracic pedicle screw placement compared to the free-hand technique. However, both techniques were found to achieve satisfactory safety of screw insertion in scoliotic patients.
Comments and Questions:
1. CSF leakage during scoliosis surgery can be a serious complication, which may be resulted from violence into nerve tissue and cause terrible neurologic deficits. Could you provide more clinical information about the two cases of CSF leakage during scoliosis surgery and explain the potential reasons why this complication occurred specifically in the 3D navigation template group and not in the free-hand technique group?
2. You may add footnotes to the tables in order to provide clear definitions for the classifications used in them, such as vertebral rotation and screw position, even if they have already been defined in the text.
3. The small sample size limited the accuracy. If the sample number could get more, it would be more powerful to convince readers.
4. I believe that the screw accuracy using the 3D template technique should have been superior to the free-hand technique in more rotatory vertebrae, but your study was not able to demonstrate this. Besides the small sample size, what other factors do you think may have contributed to the lack of significant difference in screw accuracy between the 3D template and free-hand techniques in more rotatory vertebrae in your study?
Author Response
Thank you vey much for your comments. Please see the attachment.

Reviewer 2 Report
Thank you for submitting your manuscript on the use of 3D templates to improve accuracy of insertion of thoracic pedicle screws. It is an interesting and timely article. My institution also uses a similar technique. We now have experience with over 75 cases. My questions, concerns and suggestions include:
Introduction
(1). Page 1, Lines 36 and 41. The terms "Cobb angle" and "gold standard" are jargon and unacceptable in scientific writing. Cobb is not an angle but rather a technique used to measure radiographic spinal angle. Cobb angle is a common term but "major coronal curve" or just "major curve" is the appropriate term. Please see Scoliosis Research Society (SRS) Terminology for further clarification. Gold standard is also inappropriate and should be replaced by a term that implies it is the "best current technique".
(2). Page 1, Line 45. I am not aware that screw malposition occurs as frequently as described in these references. These are not particularly recent references (1996-2017). Usually, it is about 5%. Please change to more recent references provide more detail.
(3). Page 2, Line 63. Are you referring to the entire thoracic spine or just the area of the major coronal curve. This is unclear and needs to be clarified. The apical vertebrae rotate opposite the direction of the major curve. This results in both the curvature and the associated rib hump.
Materials and Methods
(4). Page 2, Line 92. I have a major concern with your inclusion criteria. In a small series of patients as you have the diagnoses need to be the same for accurate comparisons. It appears that you had 9 cases of idiopathic scoliosis, and 3 cases each of congenital scoliosis and degenerative scoliosis between your two groups. An outlier could bias your results in either group. Also, what did you mean by degenerative scoliosis? This is not a common classification of scoliosis.
(5). Page 3, Line 99. What was your radiation dose for your CT scans? Was it standard or reduced? If reduced what was the percentage compared to the standard dose? This is an important question because of the risk of radiation related sarcomas in early to mid-adult life.
(6). Page 3, Line 113. Our templates clearly define the starting or entry point as well as the direction, screw diameter, and screw length. This allows the path and depth of drilling. The tract can be quickly felt, tapped, re-felt , and the screw inserted with marked accuracy. This saves considerable time.
(7). Page 3, Line 120. What does Error! Reference source not found mean? You used this frequently in your manuscript.. This is not appropriate for a scientific manuscript.
(8). Page 4, Line 149. Anteroposterior and lateral fluoroscopy images should not be necessary after insertion of each screw. Final check of the instrumented construct can be made at completion provided there are no concerns. This will further decrease or limit radiation exposure.
(9). Page 4, Line 159. Another CT scan! This again increase radiation exposure. Was this a standard or reduced dose? Please make a comment on the total amount radiation each patient received using your procedure.
Results
(10). Page 5, Line 192. Please use only one decimal place for your length measurements. Two decimal places is not clinically relevant. Make this change throughout your text and any tables or figure legends where they were used.
(11). Page 6, Line 200. Please round off your spinal angle measurements to whole numbers. The standard error of measurement for spinal radiographs is 3-5 degrees. Thus, decimal places do not add accuracy or significance.
(12). Page 7, Line 237. Your screw malposition data is of significant concern! This indicates numerous malposition's per patient. This is very unusual in templated insertions. How many screws needed to be repositioned at surgery or later? In my free-hand experience this is uncommon.
(13). Page 7, Line 250. In Table 6 how do you explain essentially the same operative time but significantly greater intraoperative blood loss? Usually time is directly related to blood loss. Also, why the prolonged length of stay in both groups?
(14). Do you have data for the costs associated with procurement and use of the system used to manufacture your templates? This is important information.
Conclusion
(15). Page 9, Line 322. I am uncertain if can truly make any significant conclusions based on the limited number of patients in the study groups. We have documented decreased operative time (almost one hour), decreased intraoperative blood loss, and increased accuracy. This comes at increased cost. As such I agree with your recommendations for an RCT.
The English is satisfactory. I have some questions regarding their use of certain terms and their scientific English. These were addressed sequentially in the previous section.
Reviewer 3 Report
This paper concerns very important problem for spine surgeons: quality of screw placement. The paper is prepared with a lot of data, however:
1. in introduction I did not find anything about a unique goal of the paper (something new, that should be find/reveal/discover)
2. in the text, I could not find info about intraoperative radiation exposure time and dose in 3D template and free-hand group.
3. what's new in this research comparing with cited papers (21,42) - It is not clearly visible.
Author Response

(The authors gave the same response as above.)

Reviewer 4 Report
Title: May be more clearer
Abstract: Very generalised. Please present the important numerical details/ data. No control group details mentioned. The conclusion cannot be correct unless the results are compared with control group or literature data
Intro: Pls elaborate the literature evidence on 3D navigation template
Methods: Reasonably well explained
Results: Error! Reference source not found.
What does this mean?
Results overall presented well
Discussion: Pls elaborate relevant studies in the literature. Compare the findings with the current study in terms of accuracy
Conclusion: Very long. Can be precise
Reasonable quality of language
Round 2
Reviewer 1 Report
It has been modified well, and there are no special comments
Author Response
Thank you very much for your confirmation and kindness.
Reviewer 2 Report
Thank you for revising your manuscript. Unfortunately, it has not been significantly improved. In fact, many of your responses to my questions and concerns were unsatisfactory. I remain concerned that your results of this study are not current with other centers using this innovative technique. In my program We have already performed approximately 75 cases (Firefly - Mighty Oaks Medical, Englewood, CO) with significantly different results. As a consequence, I remain concerned about proceeding with publication. Using your sequential responses to my previous numbered questions I have the following comments to your answers.
(1). I appreciate your changes regarding the terms "Cobb angle" and "gold standard". The new terms are much more appropriate.
(2). You have not satisfactory addressed this question. I tried to review your new references but not all could be searched on PubMed. I continue to feel that you are not adequately addressing my initial concern. Malposition is very different from those that are satisfactorily inserted but not in ideal based on current classification systems. Most surgeons are concerned with those screws that require intraoperative or postoperative revision because of a medial breach or being too long. Also, your cited references were primarily from Asian institutions. Was there a reason?
(3). Thank you for your clarification.
(4). I strongly disagree on your patient groups. It is important that they have comparable diagnoses. Mixed diagnoses typically yield mixed results. Only your idiopathic scoliosis patients were truly comparable. I also noted that your study patients were predominantly adults. It important that adult and pediatric patients be separated as it is well known they have different technical issues and results. For these reasons I do not feel your results can be based only by technique (templated and free-hand insertions).
(5). This is not a satisfactory response to my question. The correct response would have been to simply state that the standard radiation dose was used. You could then mention in your discussion that going forward reduced radiation exposure to the level necessary for satisfactory interpretation should be requested and used.
(6). Not recording screw insertion time is hard to understand as decreased operative time is a major advantage of templates. You cannot subsequently make your statement on page 9 that "total surgical time in the template group was less than in the free-hand group, no significant differences were observed". We have found that our operative times are reduced by approximately one hour.
(7). Thank you for clarifying this point of confusion.
(8). Your response is not clear. While it is obvious that not every screw requires imaging after insertion those without concerns can be evaluated as a group with AP or PA and lateral images. This will decrease the radiation exposure for your patients. This is another advantage of the use of templates.
(9). This response is still unclear, Please simply state, if true, that postoperative CT scans are obtained. In the Discussion you need to mention the importance of decreasing total radiation exposure because of concerns of later radiation induced malignancies. Please include the appropriate references for these concerns.
(10). Thank you for correcting your decimal places in measured lengths.
(11). Thank you for rounding your mean radiographic spinal angles. This makes them easier to comprehend. I also noticed this was inconsistent and that you failed to round up your percentages to whole numbers, particularly in the Discussion.
(12). Your responses to this question was unsatisfactory. The answer should be simple. How many screws had breaches (medial or lateral)? How many were too long? How many required intraoperative and postoperative revision. In our template experience this is very low. Your three points are excuses and unacceptable.
(13). Your responses to essentially the same operative time but significantly greater blood loss in the free-hand group is also unacceptable. It is also the exact opposite of our experience. Templates increase insertion speed and accuracy and decrease perioperative blood loss. Any information on this in the literature? The fact that your two study groups were not comparable by diagnoses, age at and other factors contributed to these results and significantly flawed your results.
(14). There was a cost, and in fact, they are relatively expensive. It would be appropriate to speculate the costs. This will be important to our readers.
(15). I continue to agree with a larger RCT. You will still need appropriate, comparable categories to determine any significant results.
Reviewer 3 Report
The paper has been imp[roved.
In my opinion, this version may be published.
Author Response

(The authors gave the same response as above.)

Reviewer 4 Report
The recommended changes have been added to the manuscript. The manuscript may be accepted in the current format
Well written overall
Author Response

(The authors gave the same response as above.)
